# A Hybrid MCDM Model for Evaluating Strategic Alliance Partners in the Green Biopharmaceutical Industry

**Mu-Hsin Chang [1], James J. H. Liou [2],[\*] and Huai-Wei Lo [1]**

[1] Graduate Institute of Industrial and Business Management, National Taipei University of Technology, 1, Sec. 3, Zhongxiao E. Rd., Taipei 10608, Taiwan

[2] Department of Industrial Engineering and Management, National Taipei University of Technology, 1, Sec. 3, Zhongxiao E. Rd., Taipei 10608, Taiwan

[\*] Correspondence:jamesjhliou@gmail.com

**Abstract:** Since the rise of strategic alliances which play such an important role in industry today, the biopharmaceutical industry worldwide has entered an era of rapid change and collaborative thinking. The strategic alliance is one of the most important strategies for the green biopharmaceutical industry. Member organizations in these alliances work together to create more advantageous biotechnologies based on environmental protection to achieve mutual benefits. In the past, there have been only a few studies discussing partner evaluations and the selection process for the green biopharmaceutical industry, so the criteria or indicators are still not complete. Therefore, this study proposes a novel multi-criteria decision-making (MCDM) framework for strategic alliance partner evaluation that combines the best-worst method (BWM) and the fuzzy TOPSIS technique based on the concept of aspiration level (called fuzzy TOPSIS-AL) to evaluate the performance and priority rankings of strategic alliance partners. The BWM overcomes the shortcomings of small sample sizes and streamlines the number of conventional pairwise comparisons needed. The fuzzy TOPSIS-AL technique introduces the concept of the aspiration level, thereby leading to more reasonable suggestions for improvement. In addition, data from a multinational green biopharmaceutical company survey are utilized to demonstrate the validity and applicability of the proposed model.

**Keywords:** MCDM; BWM; fuzzy TOPSIS; aspiration level; strategic alliance partner; biopharmaceutical

## 1. Introduction

Extreme changes in climate and frequent natural disasters have forced governments to pay more attention to environmental protection and have enacted many environmental protection regulations and penalties. In recent years, international governments have developed a number of environmental policies for energy-intensive industries (EIIs), hoping to reduce anthropogenic greenhouse gases emissions through legislation. EIIs include industries such as electronics, chemicals, machinery, petroleum, automotive, and biotechnology. They emit more than 45% of all industries and public activities [1]. For global sustainable development, the World Trade Organization (WTO), the World Health Organization (WHO), the European Union (EU), and other international organizations have enacted many environmental protection monitoring legislations and agreements [2].

The biopharmaceutical industry is a relatively new energy-intensive industry that is recognized as one of the most promising industries in the 21st century, and its development is critical to the technological advancement of global healthcare [1,3]. Emerging biotech pharmaceuticals are made up of complex biomolecules that provide solutions for chronic and debilitating diseases [4]. Biotech products have been approved for marketing in Europe, and the market value of these products is expected to reach US$35 billion in 2020 [5]. The biopharmaceutical industry uses bio-based products to make commercially valuable drugs, including hormones, fusion proteins, cytokines, blood factors, vaccines, and redox molecules [6]. The biopharmaceutical industry strives to meet the rigorous standards required for the production of therapeutic drugs through a series of complicated manufacturing processes and costly clinical trials. To minimize investment costs, many companies look for partners or ways to outsource. Because of the requirements of advanced technology, high investment, and long-term R&D cycles, the pharmaceuticals industry is classified as a high-risk industry [7]. Therefore, it is especially advantageous for biopharmaceutical companies to form strategic alliances with other companies upstream and downstream the supply chain to enhance competitiveness, including shortening product development time, reducing development costs and risks, and increasing product diversity.

The goal of a strategic alliance is to integrate two or more companies, and the joint management of the overall supply chain can achieve resource sharing and market diversification. In general, strategic alliances involve formal legal or private informal partnerships, and partners can complement their strengths and weaknesses to reduce business risk [8]. The biggest advantage of strategic alliances in the biopharmaceutical industry is the ability to jointly develop more valuable biotech products and promote the development of human health care [9]. Due to the rise of environmental awareness, governments in different countries have established environmental regulations for the biopharmaceutical industry through legislative units.

At present, the most common method of strategic alliance partner evaluation in the biopharmaceutical industry is financial cost benefit analysis, which focuses on the profit and loss balance of business operations, that is, financial and cost indicators, ignoring the goal of environmental protection [10,11]. According to the literature review, the strategy for partner selection for biopharmaceutical production still focuses on financial performance [12,13]. In addition, according to Ramasamy et al. [1], environmental standards are not talked about in the research related to the biopharmaceutical industry. Green biopharmaceutical production is a modern production model that takes into account environmental impacts and resource efficiency. The goal is to minimize the negative impact of pharmaceutical production on the environment. The evaluation criteria for green biopharmaceutical production should include procedures along the supply chain from product design and manufacturing to transportation and scrapping. Compared to other industries, there has been relatively little research on the evaluation of green biopharmaceutical strategic alliance partners. The first multi-criteria decision-making (MCDM) framework for the bio-manufacturing industry was developed by George et al. [3], whose evaluation criteria included earnings capacity, asset utilization, long-term solvency, productivity, and manufacturing knowledge. Shakeri and Radfar [14] presented a comprehensive model for performance evaluation of the biopharmaceutical industry strategic alliance. They mainly surveyed the strategic performance of alliances between manufacturers and exporters of medical biotechnology products in Iran between 2000 and 2012. The model explores the relationship between several factors, including partner fit, alliance ability, capital amount, and learning ability. In recent years, strategic alliances of biopharmaceutical multinationals have also received much attention, especially in the context of cultural diversity in research and development and innovation [15]. Unfortunately, there is still no research to establish a complete strategic alliance partner evaluation framework for the green biopharmaceutical industry.

Some advanced countries have listed the biopharmaceutical industry as one of the key development projects. Therefore, evaluating strategic alliance partners in the green biotechnology industry is an important task. The MCDM method has excellent evaluation performance in complex environments. It does not require the basic assumptions of traditional statistics, and only requires a small sample of expert interview data. The MCDM's goal is to integrate objective survey data with expert subjective judgments and provide effective management information to support decision-makers in developing best strategies [16]. Common methods for determining weights are the analytic hierarchy process (AHP) [17], analytic network process (ANP) [18], best-worst method (BWM) [19], decision making trial and evaluation laboratory (DEMATEL) [20], and entropy [21]. Methods for performance integration and evaluation include technique for order preference by similarity to an ideal solution (TOPSIS) [22], Visekriterijumska Optimizacija i Kompromisno Resenje (VIKOR) [23], ELECTRE [18], preference ranking organization method for enrichment evaluation (PROMETHEE) [24], and data envelopment analysis (DEA) [25]. MCDM methods have been widely used in the assessment and selection of various industries. Büyüközkan et al. [26] used AHP and TOPSIS to determine the ranking of partners in the logistics value chain. The criteria for evaluation are mainly divided into the individual ability of the partner and the organizational cooperation ability of the alliance. Wang et al. [25] proposed a selection framework for aerospace and defense alliance partners with the use of the DEA approach to predict the future operational performance of viable alliance partners. There are also some studies that use MCDM to evaluate strategic alliance partners, such as collaborative development of communities [27], cocreation strategies for telecom operators [28], and innovation and entrepreneurship of clean technologies [29].

This paper proposes a strategic alliance partner evaluation framework for the green biotechnology industry, using a hybrid MCDM approach to evaluate partners' performance. First, based on the relevant literature, and the discussion with the decision-makers of the target company is made to establish a complete evaluation criteria system, especially the environmental protection criteria. Second, the BWM method is used to obtain the weight of the criteria. The BWM method is one of the most popular weight calculation methods in the past five years. It overcomes the two shortcomings of AHP, that is, the large number of pairwise comparisons and the poor consistency. Finally, modified fuzzy TOPSIS is used to calculate the total evaluation scores of each partner. The addition of fuzzy theory overcomes the problem of information uncertainty. In addition, this paper improves the TOPSIS technique proposed by Kuo [30] and introduces the concept of the aspiration level into the calculation process of TOPSIS, thereby avoiding having to select the best apple from a barrel of rotten apples [31–33]. The improved TOPSIS can be used to obtain the room for improvement for each partner based on their distance from the aspiration level, so that more management information can be obtained in practical applications. Finally, this study applies data obtained from the survey of a multinational green biopharmaceutical company in Taiwan as a case study. The method can help decision-makers be more systematic in the decision-making process and the results provide more reliable suggestions for improvement of their partners.

The rest of the paper is organized as follows. Section 2 presents the criteria for evaluation of green biopharmaceutical industry strategic alliance partners. Section 3 describes the proposed hybrid MCDM model approach and its basic concepts. Section 4 demonstrates the feasibility and practicality of the proposed model in a real-world application. Section 5 summarizes the discussion of the whole study and provides future research directions.

## 2. Literature Review for Strategic Alliance Partner Evaluation Criteria

Partner evaluation criteria are very important for the performance evaluation of partners in a green biopharmaceutical industry strategic alliance. First, the most important criteria should be fully integrated into the evaluation system to reflect the characteristics and implications of the strategic alliance. The initial criteria are extracted from a review of the relevant academic literature and expert interviews. Second, experts in business management, economic and social development,

and environmental protection are gathered to form a decision-making group. The group, including both academics and business practitioners, reviews the initial criteria and selects the most essential criteria. Finally, the Delphi method is used to integrate the experts' opinions, to determine the final evaluation criteria. A strategic alliance partner evaluation framework for green biopharmaceutical companies is identified. The main structure consists of five dimensions, namely, economic resources, innovation capability, organizational management, risk factor, and environmental protection. Moreover, these five dimensions contain several subcriteria, and a total of 25 evaluation criteria are included in the evaluation framework. The framework of the strategic alliance partner evaluation criteria for green biopharmaceutical companies is shown in Figure 1. In a strategic alliance for the biopharmaceutical industry, more emphasis is placed on core technical patents and R & D capability than would be the case in other manufacturing industries. Due to the long development time and high level of risk of biopharmaceutical products, partners must have the necessary R & D capability to collaborate with the strategic alliance. In fact, the more invention patents held within the alliance, the stronger the overall competitiveness. In addition, in order to comply with the standards of environmental protection, many biopharmaceutical companies have moved towards green manufacturing. The green criteria developed in this study can be used to determine whether prospective partners value green development. Elia et al. [15], Wang et al. [25] and Büyüközkan et al. [26] mentioned that strategic alliances in various industries must consider financial resources, organizational management and risks. Our evaluation framework includes their ideas and opinions.

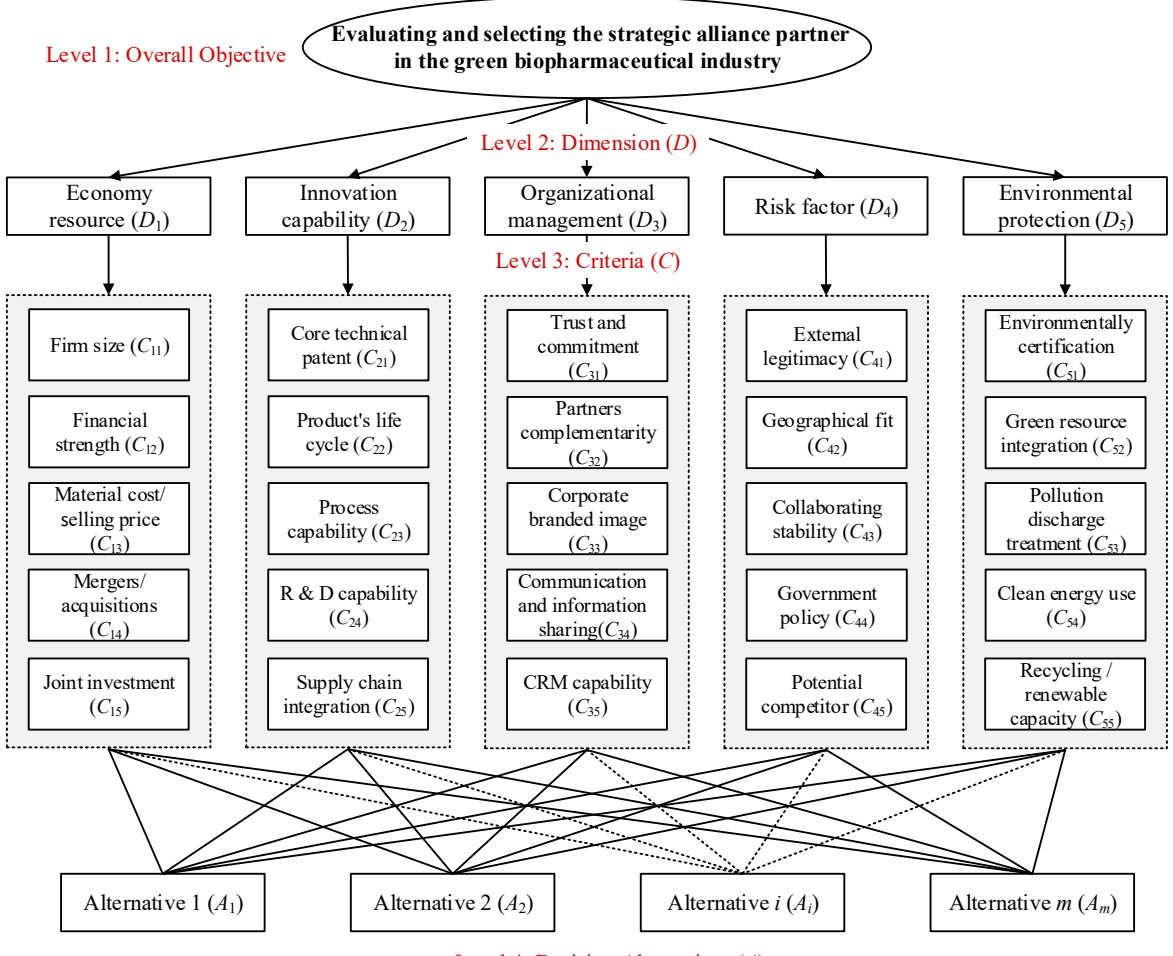

**Figure 1.** The framework of green biopharmaceutical industry strategic alliance partner evaluation criteria.

### 2.1. Economy Resource

Economic resources ($D_1$) is an evaluation dimension of a partner's business performance and management capabilities, including firm size ($C_{11}$), financial strength ($C_{12}$), material cost/ selling price ($C_{13}$), mergers/ acquisitions ($C_{14}$), and joint investment ($C_{15}$). Firm size ($C_{11}$) is to evaluate a company's capital, turnover, number of employees, market share, and even corporate management and structure. These elements are combined to represent the size and internationalization level of the firm [34–36]. Financial strength ($C_{12}$) is an assessment of the company's financial statements of assets and liabilities, income statement and cash flow. Information on corporate solvency, company internal control, board functions, business conditions, etc. are all important indicators of financial capabilities [36,37].

Material cost/selling price ($C_{13}$) is an important indicator for assessing the company's profit, including all direct and indirect materials, working hours, equipment, plant, fixed and variable costs of operating and sales, and net/gross profits. The criteria can reflect the firm's pricing and cost control capabilities [22,38]. Mergers/acquisitions ($C_{14}$) are the means of cooperation between strategic alliance partners to assess whether the company has the value of being acquired. This indicator is a common indicator for strategic alliances. Larger companies often want to increase supply chain management capabilities through acquisitions [39,40]. Joint investment ($C_{15}$) refers to the willingness of partners to invest their respective superior resources in the joint venture to achieve co-financing, share risks and share benefits. This criterion is very important in the biopharmaceutical industry because it reflects the sincerity and attitude of the partners [40,41].

### 2.2. Innovation Capability

Innovation capability ($D_2$) is one of the most important competencies in the biopharmaceutical industry, covering all product development, including core technical patent ($C_{21}$), product's life cycle ($C_{22}$), process capability ($C_{23}$), R & D capability ($C_{24}$), and supply chain integration ($C_{25}$). Core technical patent ($C_{21}$) is one of the conditions for a company's competitive advantage. The strategy for developing patents is to develop new markets to ensure competitive advantage. The source of patents will not be limited to self-development, but will also include the purchase of patents, mergers and acquisitions of other companies, and technology licensing [38,42,43]. Product's life cycle ($C_{22}$) refers to the time course of the product through the introduction period, growth period, maturity period and recession period. In general, companies expect the life cycle of biopharmaceutical products to be as long as possible, indicating that the product has a long-term contribution to health care [7].

Process capability ($C_{23}$) is an assessment of whether a process has a quality that meets the customer's requirements under fixed production conditions and stable controls [15,22,38]. R&D capability ($C_{24}$) refers to whether the company has mastered the leading technology and knowledge based on R&D, clearly understands the needs of the market, and has the ability to innovate products [7,22]. Supply chain integration ($C_{25}$) refers to the strategic alliance's supply chain integration capabilities, including the process of feeding, production, inventory and sales of all products. The role of each strategic alliance partner is both the supplier and the customer, so each partner's positioning in the overall supply chain is important [22,36].

### 2.3. Organizational Management

Organizational management ($D_3$) is to understand the management capabilities of partners for internal and external organizations. The criteria for evaluation include trust and commitment ($C_{31}$), partner complementarity ($C_{32}$), corporate branded image ($C_{33}$), communication and information sharing ($C_{34}$), and customer relationship management (CRM) capability ($C_{35}$). Trust and commitment ($C_{31}$) is the foundation for the stability of strategic alliance cohesion. Trust and termination of punishment can be an effective tool to motivate commitments and improve the effectiveness of the alliance. This criterion is considered by many industries to be one of the key factors for the success of a strategic alliance [7,13]. Partner complementarity ($C_{32}$) refers to the integration and management of different

resources, capabilities, and technologies that partners have to enhance the competitiveness of strategic alliances. The heterogeneous nature of the partner's industry maximizes complementarity [7,36].

Corporate branded image ($C_{33}$) means the society's perception and impression of the company or product, fully reflecting the value of the company. The company's excellent brand image enhances profit and expansion channels, and thus increases product market share [4,7]. Communication and information sharing ($C_{34}$) is one of the most important factors in the success of a strategic alliance. If the message can be announced instantly and correctly on a common information platform, partners can share information and quickly grasp the information of the alliance. This criterion is used to assess the capability of partners to manage information systems [4,8]. CRM capability ($C_{35}$) refers to the level of service that an enterprise meets the needs of its customers. It mainly uses high-performance information technology to collect data and analyze customer needs, and quickly process customer orders. The ability of corporate CRM is highly correlated with loyalty [4,7].

### 2.4. Risk Factors

The biopharmaceutical industry is one of the high-risk industries, so risk management of strategic alliances is an important task. The risk factor consists of five criteria, namely external legitimacy ($C_{41}$), geographical fit ($C_{42}$), collaborating stability ($C_{43}$), government policy ($C_{44}$), and potential competitor ($C_{45}$). External legitimacy ($C_{41}$) refers to the fact that enterprises and alliance partners must first obtain the legalization of the local government in the activities of multinational economic organizations before they can start operations and trade, and must avoid legal criminal responsibility [44]. Geographical fit ($C_{42}$) refers to the need to assess cultural differences and business values of the region in advance when the company organizes economic activities in different regions to avoid losses and reputational damage [28,38].

Collaborating stability ($C_{43}$) is the attempt of companies to achieve the common goal of strategic alliances, and partners are built on mutual trust. Good cooperation stability effectively improves the performance of the overall supply chain and accelerates the expansion of new markets [28]. Government policy ($C_{44}$) refers to that the partners must comply with local government policies, laws, and related regulations. Generally speaking, the government's formulation of relevant business regulations will significantly affect the decision-making and management policies of the company [2,45]. Potential competitor ($C_{45}$) refers to a competitor that does not pose a conflict of interest to the company for the time being, but may pose a certain degree of threat to the interests of the company in the future. Partners should have timely intelligence and information to assess potential competitors that would otherwise pose a threat to the company and even be eliminated by the market [46].

### 2.5. Environmental Protection

For the first time, this paper proposes environmental protection assessment criteria for the biopharmaceutical industry, including environmentally certification ($C_{51}$), green resource integration ($C_{52}$), pollution discharge treatment ($C_{53}$), clean energy use ($C_{54}$) and recycling/renewable capacity ($C_{55}$). Environmentally certification ($C_{51}$) means that the alliance partners must comply with local government environmental regulations and obtain relevant environmental certifications and certificates. This indicator has become an indispensable condition for green supplier and partner evaluation [2,47]. Green resource integration ($C_{52}$) means that companies must respect the natural environment and protect the ecology in the production process to create a green supply chain system. Strategic alliance partners value the integration of green resources in the supply chain, which can effectively enhance corporate image and implement environmental protection policies [2,47].

Pollution discharge treatment ($C_{53}$) is an environmental policy that assesses whether a company is actively implementing reduction of pollutant emissions, thereby improving the energy efficiency of enterprises. Polluting emissions from the biopharmaceutical industry, especially chemical testing, can seriously affect the environment [2,47]. Clean energy use ($C_{54}$) means that the company is committed to developing clean, efficient, and systematic technologies for the production of biopharmaceuticals, and promoting the use of environmentally friendly energy. Clean energy includes natural renewable energy such as water, wind, tides, and solar energy [2,22,47]. Recycling/renewable capacity ($C_{55}$) refers to the ability of enterprises to pay attention to the recovery and regeneration of materials, consumables, or energy in the process of R&D and production. When the company has the ability to recycle and regenerate, it can not only reduce production costs, but also reduce environmental damage [2,22,47].

## 3. The Proposed Model for Strategic Alliance Partner Evaluation

This section describes the MCDM methods used and their detailed calculation processes, including best worst method, fuzzy set theory, and fuzzy modified TOPSIS-AL technique.

### 3.1. The Best-Worst Method

BWM was proposed by Rezaei [19]. Compared with AHP, the BWM questionnaire is more concise and achieves better consistency. BWM has been widely used in decision making in various industries. Rezaei et al. [48] proposed the service quality (SERVQUAL) model to assess the service quality of the aerospace baggage handling system and investigated passengers from different nationalities. Through the analysis of BWM, it was determined that "reliability" is the most important indicator. Omrani et al. [49] combined the BWM and MULTIMOROA methods to assess the human development index. This study demonstrates that BWM is a more efficient method of calculating weights than AHP. There are other practical applications, such as site selection [50], supplier evaluation [51], company performance evaluation [52], key factors analysis for sustainable building [53], and so on. The detailed processes of BWM obtaining weights are described as follows:

*Step 1. Determine the evaluation criteria set of the decision system*

Decision-makers develop a set of criteria $\{c_1, c_2, \ldots, c_n\}$ for evaluating the strategic alliance partners.

*Step 2. Select the best and worst criteria*

Based on the $n$ criteria developed in Step 1, decision-makers pick the best (i.e., most satisfied, most preferred, or most important) and the worst (i.e., least satisfied, least preferred, or least important) criteria. The best and worst criteria chosen are key factors influencing the results of the BWM analysis.

*Step 3. Compare the best criterion with other criteria to generate BO (Best-to-Others) vector*

Decision-makers assess the relative importance level of the best criteria and other criteria, as shown in Table 1. The evaluation scale ranges from 1 to 9 and the BO vector is generated. Scale 1 is considered to be equally important, and scale 9 is absolutely important and belongs to the highest level of scale. It is expressed as:

$$A_{Bj} = (a_{B1}, a_{B2}, \ldots, a_{Bn})$$

where $a_{Bj}$ indicates the importance level of the best criterion $B$ relative to the criterion $j$, and the comparison between the best criterion and itself must be 1 (i.e., $a_{BB} = 1$).

**Table 1.** Evaluation scales of the BWM questionnaire.

| Linguistic Variables | Code |
| --- | --- |
| Equally important | 1 |
| Moderately more important | 3 |
| Strongly more important | 5 |
| Very strongly more important | 7 |
| Extremely more important | 9 |
| Intermediate values | 2, 4, 6, 8 |

*Step 4. Compare the worst criterion with the other criteria and generate OW (Others-to-Worst) vector*

Similar to Step 3, the decision-makers evaluate the relative importance level of other criteria to the worst criterion, and then produces an OW vector, which is expressed as:

$$A_{jW} = (a_{1W}, a_{2W}, \ldots, a_{nW})^T$$

where $a_{jW}$ indicates the importance level of the remaining criterion $j$ relative to the worst criterion $W$, and the comparison between the worst criterion and itself must be 1 (i.e., $a_{WW} = 1$).

*Step 5. Calculate the optimal weights $\left(w_1^*, w_2^*, \ldots, w_n^*\right)$ for each criterion*

The best criterion weight value is obtained by the linear programming (LP) model. The input data is BO and OW vectors (the weight ratio of the best criterion to the remaining criteria and the weight ratio of the remaining criteria to the worst criterion). We should find a solution where the maximum absolute differences $\left|\frac{W_B}{W_j} - a_{Bj}\right|$ and $\left|\frac{W_j}{W_W} - a_{jW}\right|$ for all $j$ is minimized. Considering the non-negativity and sum condition for the weights. The complete model is expressed as follows:

$$min \; max_j \left\{\left|\frac{W_B}{W_j} - a_{Bj}\right|, \left|\frac{W_j}{W_W} - a_{jW}\right|\right\};$$

$$s.t. \begin{cases} \sum_j w_j = 1; \\ w_j \geq 0, \text{for all } j. \end{cases} \tag{1}$$

In Equation (1), the objective function of the minimized maximum can be converted to a minimized objective function for calculation. The minimized objective function after conversion can be presented by the following model:

$$min \; \xi;$$

$$s.t. \begin{cases} \left|\frac{W_B}{W_j} - a_{Bj}\right| \leq \xi; \\ \left|\frac{W_j}{W_W} - a_{jW}\right| \leq \xi; \\ \sum_j w_j = 1; \\ w_j \geq 0, \text{for all } j. \end{cases} \tag{2}$$

Equation (2) has the possibility to generate multiple optimal solutions. Therefore, Rezaei [54] proposed a linear BWM model and modified the minimized objective function as:

$$min \; \xi^L;$$

$$s.t. \begin{cases} \left|w_B - a_{Bj}w_j\right| \leq \xi^L; \\ \left|w_j - a_{jW}w_W\right| \leq \xi^L; \\ \sum_j w_j = 1; \\ w_j \geq 0, \text{for all } j. \end{cases} \tag{3}$$

Equation (3) is a linear problem, it only gets a single optimal solution, and the best weight value $\left(w_1^*, w_2^*, \ldots, w_n^*\right)$ is obtained. $\xi^L$ can be regarded as a consistency indicator, and when it is close to 0, it means that it has a high degree of consistency.

## 3.2. The Fuzzy Modified TOPSIS-AL Technique

TOPSIS is currently one of the most effective MCDM methods for integrating performance values. The method mainly finds positive and negative ideal solutions in the alternative combinations, and determines the relative position of each alternative by calculating the distances between each alternative and the positive and negative ideal solutions. The best alternative is to be closest to the positive ideal solution and farthest away from the negative ideal solution. TOPSIS is easy to understand and operate and has been used in many problems [55–58]. Furthermore, when performing

decision-making processes in an uncertain context, they are often influenced by subjective and vague judgments. Therefore, Zadeh [59] first introduced fuzzy set theory as a soft computing method for decision ambiguity. According to the definition of fuzzy sets, expert opinions are usually described by linguistic variables. In practice, linguistic variables can be represented by fuzzy numbers, forming a set of ambiguities. The most common linguistic variable is the triangular fuzzy number (TFN), proposed by Pedrycz [60]. This paper combines TOPSIS with fuzzy theory to reflect the inaccuracy of the practice assessment environment and to replace the relatively better solution in the existing solutions with the aspiration level. The detailed TOPSIS operation steps are described as follows.

*Step 1. Define the symbol*

Suppose there are $m$ alternatives $A_i = \{A_1, A_2, \ldots, A_m\}$, $n$ criteria $c_j = \{c_1, c_2, \ldots, c_n\}$, and the weight of the criteria is defined as $w_j = \{w_1, w_2, \ldots, w_n\}$. Each expert $D_k$ ($k = 1, 2, \ldots, p$) evaluates the performance of the alternative $A_i$ ($i = 1, 2, \ldots, m$) according to the criterion $c_j$ ($j = 1, 2, \ldots, n$). Table 2 shows the scales of performance evaluation.

**Table 2.** Linguistic variables and corresponding triangular fuzzy numbers.

| Linguistic Variables | Code | Fuzzy Numbers |
|:---:|:---:|:---:|
| Very poor | VP | (0, 1, 2) |
| Poor | P | (2, 3, 4) |
| Fair | F | (4, 5, 6) |
| Good | G | (6, 7, 8) |
| Very good | VG | (8, 9, 10) |

*Step 2. Construct an initial fuzzy decision matrix $\widetilde{X}$*

Expert $D_k$ evaluates all alternatives according to the scales of Table 2. This paper uses the arithmetic mean to summarize the evaluation values of all experts and obtains the initial evaluation fuzzy decision matrix, expressed as

$$\widetilde{X} = \left[\widetilde{x}_{ij}\right]_{m \times n} = \begin{bmatrix} \widetilde{x}_{11} & \widetilde{x}_{12} & \cdots & \widetilde{x}_{1j} & \cdots & \widetilde{x}_{1n} \\ \widetilde{x}_{21} & \widetilde{x}_{22} & \cdots & \widetilde{x}_{2j} & \cdots & \widetilde{x}_{2n} \\ \vdots & \vdots & \ddots & \vdots & \ddots & \vdots \\ \widetilde{x}_{i1} & \widetilde{x}_{i2} & \cdots & \widetilde{x}_{ij} & \cdots & \widetilde{x}_{in} \\ \vdots & \vdots & \ddots & \vdots & \ddots & \vdots \\ \widetilde{x}_{m1} & \widetilde{x}_{m2} & \cdots & \widetilde{x}_{mj} & \cdots & \widetilde{x}_{mn} \end{bmatrix} \quad (4)$$

$$\widetilde{x}_{ij} = \left(x_{ij}^l, x_{ij}^m, x_{ij}^u\right), \ i = 1, 2, \ldots, m, \ j = 1, 2, \ldots, n$$

where $x_{ij}^l = \frac{1}{p}\sum_{k=1}^{p} x_{ijk}^l$, $x_{ij}^m = \frac{1}{p}\sum_{k=1}^{p} x_{ijk}^m$ and $x_{ij}^u = \frac{1}{p}\sum_{k=1}^{p} x_{ijk}^u$, $k = 1, 2, \ldots, p$.

*Step 3. Construct a normalized fuzzy decision matrix $\widetilde{X}^*$*

The purpose of normalization is to unify the units of all evaluation criteria and to make the values within the matrix bound to 0 to 1. The normalized fuzzy matrix is $\widetilde{X}^* = \left[\widetilde{x}_{ij}^*\right]_{m \times n}$. The conventional normalization method is to take the best performance value in the alternatives as the denominator, i.e.,

$$\widetilde{x}_{ij}^* = \frac{\widetilde{x}_{ij}}{max_j\{\widetilde{x}_{ij}\}} \quad (3)$$

This article introduces the concept of the aspiration level into this step. The modified formula is

$$\widetilde{x}_{ij}^* = \frac{\widetilde{x}_{ij}}{x^{aspire}} \quad (6)$$

where $x^{asprie} = 10$ (the highest level of the evaluation scales).

*Step 4. Construct the weighted normalized fuzzy decision matrix $\widetilde{X}^{**}$*

Considering the importance of each criterion, the weighted value ($w_j$) of the criterion evaluation is multiplied by the normalized fuzzy decision matrix $\widetilde{X}^*$ to obtain the weighted normalized fuzzy decision matrix. The calculation method is as follows.

$$\widetilde{X}^{**} = \left[\widetilde{x}_{ij}^{**}\right]_{m \times n} = \widetilde{x}_{ij}^* \cdot w_j \tag{7}$$

*Step 5. Define positive ideal solutions and negative ideal solutions (PIS and NIS)*

Based on the concept of aspiration level, the positive and negative ideal solutions of the alternatives should be 1 and 0 after normalization. Therefore, the positive ideal solution and the negative ideal solution ($A^{asprie}$ and $A^{worst}$) of the alternatives are calculated as follows

$$\text{PIS} = A_j^{asprie} = (1 \cdot w_1, 1 \cdot w_2, \ldots, 1 \cdot w_n) = (w_1, w_2, \ldots, w_n) \tag{8}$$

$$\text{NIS} = A_j^{worst} = (0 \cdot w_1, 0 \cdot w_2, \ldots, 0 \cdot w_n) = (0, 0, \ldots, 0) \tag{9}$$

*Step 6. Calculate the distances between each alternative and the PIS and NIS*

Based on the definition of the Euclidean distance square, Equations (10) and (11) are used to calculate the separation distances between the alternative $i$ and the PIS and NIS. At this step, the fuzzy values have been defuzzified to be converted into crisp values.

$$d_i^* = \sum_{j=1}^n \sqrt{\frac{\left(A_j^{asprie} - x_{ij}^{**l}\right)^2 + 2 \cdot \left(A_j^{asprie} - x_{ij}^{**m}\right)^2 + \left(A_j^{asprie} - x_{ij}^{**u}\right)^2}{4}} \tag{10}$$

$$d_i^- = \sum_{j=1}^n \sqrt{\frac{\left(x_{ij}^{**l} - A_j^{worst}\right)^2 + 2 \cdot \left(x_{ij}^{**m} - A_j^{worst}\right)^2 + \left(x_{ij}^{**u} - A_j^{worst}\right)^2}{4}} \tag{11}$$

*Step 7. Calculate the closeness coefficient ($CC_i$)*

The $CC_i$ is a reliable ranking index. According to Lo et al. [22], the ranking index considers the distance between all alternatives and PIS and NIS, overcoming the shortcomings of the traditional TOPSIS ranking index. The formula is as follows:

$$CC_i = w^+ \left(\frac{d_i^-}{\sum_{i=1}^m d_i^-}\right) - w^- \left(\frac{d_i^*}{\sum_{i=1}^m d_i^*}\right), \begin{cases} -1 \leq CC_i \leq 1 \\ 0 \leq w^+ \leq 1 \\ 0 \leq w^- \leq 1 \end{cases}, i = 1, 2, \ldots, m. \tag{12}$$

Here, $w^+$ and $w^-$ represent the weights that reflect the relative importance of the PIS and NIS in the consciousness of a decision-maker, respectively. In general, both $w^+$ and $w^-$ are set to 0.5. The closer the $CC_i$ is to 1, the closer it is to the aspiration level. Conversely, when it is very close to −1, it means that the performance is extremely poor.

## 4. Illustration of Real Case

This section uses a green biopharmaceutical company as a case to illustrate the analyzed process presented in this paper.

*4.1. Problem Description*

The case company is a leading biopharmaceutical technology company in Taiwan. It is committed to the R&D, manufacturing, and sales of polymer-based biomedicine and equipment. At present, it has a number of intellectual property rights and invention patents related to green biopharmaceuticals. The company's products have successfully developed "high-viscosity tissue adhesives" and "liquid bandages", and its product quality can compete with well-known European and American manufacturers. The case company's products have been adopted by many medical centers and successfully sold in the medical retail industry worldwide. At present, the case company is more actively involved in the development of Class III (high-risk) medical advices, and signing a contract with government medical institutions, expecting to bring more profits to the enterprise. Due to the high development threshold of Class III medical advices and the high global competition and investment costs, the case company hopes to develop new products through strategic alliance partners to increase product competitiveness and market access.

At present, the case company's strategic alliance does not have a complete evaluation and management system, mainly focusing on the partners' funding and R&D capabilities. Unfortunately, environmental protection awareness has been ignored. The strategic alliance's decision-making method only determines the goals and guidelines through the discussion of the meeting. The final decision of the strategy still falls on the enterprises with the largest capital. Therefore, it is clear that it is necessary to have a complete green biopharmaceutical strategic alliance partner evaluation system and scientific analysis tools are needed to support decision making. Choosing and evaluating the right partner is an important task for business managers, and it can significantly affect a company's competitiveness.

The decision-making team consisted of eight experts from the case company, including the chairman, the chief R&D officer, the deputy general manager, the production management manager, the quality management manager, the accounting manager, and the product manager. These eight experts have more than ten years of experience in the green biopharmaceutical industry and have high ties with their partners (fixed meetings, decision discussions, and negotiations). It is necessary to evaluate the performance of partners from different professional perspectives. Through the literature review and discussion group decision-making, 25 criteria for five dimensions and their classification were identified. The evaluation model presented in this paper analyzes five strategic alliance partners.

*4.2. Obtaining the Weights of Criteria through BWM*

We applied BWM to obtain standard weights, as described in Section 3.1. First, the best and worst criteria were determined by the experts. Next, the BO and OW vectors were completed on a scale of 1 to 9. Taking the part of the dimension as an example, the best and worst dimensions selected by the eight experts are shown in Table 3. The BO vectors are shown in Table 4. Similarly, respondents were asked to evaluate other dimensions to the worst dimension. The OW vectors are shown in Table 5.

By getting the solution through Equation (3), the weight of each dimension can be determined. Following the same procedure, the weights of all criteria can be obtained. Since each expert has different backgrounds and work experience, the arithmetic mean was used to aggregate the BWM weights of the eight experts [22]. Table 6 lists the dimension weight values obtained by the eight experts via BWM calculations.

**Table 3.** The best and worst dimensions chosen by eight experts.

| Expert No. | 1 | 2 | 3 | 4 | 5 | 6 | 7 | 8 |
|---|---|---|---|---|---|---|---|---|
| Best | $D_3$ | $D_3$ | $D_2$ | $D_2$ | $D_2$ | $D_2$ | $D_2$ | $D_2$ |
| Worst | $D_5$ | $D_4$ | $D_4$ | $D_4$ | $D_5$ | $D_4$ | $D_3$ | $D_4$ |

**Table 4.** The BO vectors of the dimensions.

| Expert No. | Best | $D_1$ | $D_2$ | $D_3$ | $D_4$ | $D_5$ |
|---|---|---|---|---|---|---|
| 1 | $D_3$ | 7 | 8 | 1 | 3 | 5 |
| 2 | $D_3$ | 5 | 3 | 1 | 5 | 8 |
| 3 | $D_2$ | 3 | 1 | 5 | 7 | 7 |
| 4 | $D_2$ | 4 | 1 | 5 | 7 | 3 |
| 5 | $D_2$ | 4 | 1 | 2 | 5 | 7 |
| 6 | $D_2$ | 3 | 1 | 5 | 9 | 7 |
| 7 | $D_2$ | 3 | 1 | 6 | 4 | 2 |
| 8 | $D_2$ | 3 | 1 | 2 | 7 | 5 |

**Table 5.** The WO vectors of the dimensions.

| Expert No. | 1 | 2 | 3 | 4 | 5 | 6 | 7 | 8 |
|---|---|---|---|---|---|---|---|---|
| Worst | $D_2$ | $D_5$ | $D_4$ | $D_4$ | $D_5$ | $D_4$ | $D_3$ | $D_4$ |
| $D_1$ | 2 | 5 | 7 | 3 | 5 | 6 | 4 | 3 |
| $D_2$ | 1 | 6 | 7 | 7 | 7 | 9 | 6 | 7 |
| $D_3$ | 8 | 8 | 5 | 2 | 6 | 5 | 1 | 5 |
| $D_4$ | 4 | 2 | 1 | 1 | 2 | 1 | 3 | 1 |
| $D_5$ | 3 | 1 | 1 | 4 | 1 | 4 | 5 | 2 |

**Table 6.** Dimensional weights and average weights of the eight experts.

| Expert No. | 1 | 2 | 3 | 4 | 5 | 6 | 7 | 8 | Average |
|---|---|---|---|---|---|---|---|---|---|
| $D_1$ | 0.085 | 0.123 | 0.217 | 0.140 | 0.135 | 0.211 | 0.162 | 0.162 | 0.154 |
| $D_2$ | 0.060 | 0.205 | 0.507 | 0.498 | 38 | 0.526 | 0.416 | 0.442 | 0.387 |
| $D_3$ | 0.537 | 0.500 | 0.130 | 0.112 | 0.271 | 0.126 | 0.058 | 0.242 | 0.247 |
| $D_4$ | 0.199 | 0.123 | 0.052 | 0.062 | 0.108 | 0.047 | 0.121 | 0.057 | 0.096 |
| $D_5$ | 0.119 | 0.048 | 0.093 | 0.187 | 0.048 | 0.090 | 0.243 | 0.097 | 0.116 |

The consistency ratio (*CR*) is the reliability of the BWM questionnaire examined. The *CR* of each BWM questionnaire was less than 0.05 and the average CR was 0.023, indicating that the survey questionnaire was highly consistent [19]. Table 7 shows the combined weights of the BWM calculations. The top five criteria are R & D capability ($C_{24}$), core technical patent ($C_{21}$), corporate branded image ($C_{33}$), financial strength ($C_{12}$), and trust and commitment ($C_{31}$). Although the environmental protection criteria are not in the top five, they still affect the results of the overall evaluation system. Innovation capability ($D_2$) is the dimension the green biopharmaceutical company values most. The ranking of dimensions is Innovation capability ($D_2$) > Organizational management ($D_3$) > Economy resource ($D_1$) > Environmental protection ($D_5$) > Risk factor ($D_4$). Next, we apply the modified fuzzy TOPSIS technique to aggregate the performance data and the criteria weights for each partner.

**Table 7.** Criteria weight results.

| Dimension | Local Weight | Criteria | Local Weight | Global Weight | Rank |
|---|---|---|---|---|---|
| Economy resource ($D_1$) | 0.154 | Firm size ($C_{11}$) | 0.271 | 0.042 | 9 |
| | | Financial strength ($C_{12}$) | 0.393 | 0.061 | 4 |
| | | Material cost/ selling price ($C_{13}$) | 0.120 | 0.018 | 18 |
| | | Mergers/acquisitions ($C_{14}$) | 0.098 | 0.015 | 23 |
| | | Joint investment ($C_{15}$) | 0.119 | 0.018 | 19 |
| Innovation capability ($D_2$) | 0.387 | Core technical patent ($C_{21}$) | 0.247 | 0.096 | 2 |
| | | Product's life cycle ($C_{22}$) | 0.084 | 0.032 | 12 |
| | | Process capability ($C_{23}$) | 0.133 | 0.052 | 6 |
| | | R & D capability ($C_{24}$) | 0.428 | 0.165 | 1 |
| | | Supply chain integration ($C_{25}$) | 0.108 | 0.042 | 10 |
| Organizational management ($D_3$) | 0.247 | Trust and commitment ($C_{31}$) | 0.242 | 0.060 | 5 |
| | | Partners complementarity ($C_{32}$) | 0.175 | 0.043 | 8 |
| | | Corporate branded image ($C_{33}$) | 0.268 | 0.066 | 3 |
| | | Communication and information sharing ($C_{34}$) | 0.189 | 0.047 | 7 |
| | | CRM capability ($C_{35}$) | 0.126 | 0.031 | 13 |
| Risk factor ($D_4$) | 0.096 | External legitimacy ($C_{41}$) | 0.287 | 0.028 | 15 |
| | | Geographical fit ($C_{42}$) | 0.095 | 0.009 | 25 |
| | | Collaborating stability ($C_{43}$) | 0.166 | 0.016 | 22 |
| | | Government policy ($C_{44}$) | 0.262 | 0.025 | 16 |
| | | Potential competitor ($C_{45}$) | 0.190 | 0.018 | 20 |
| Environmental protection ($D_5$) | 0.116 | Environmentally certification ($C_{51}$) | 0.324 | 0.038 | 11 |
| | | Green resource integration ($C_{52}$) | 0.250 | 0.029 | 14 |
| | | Pollution discharge treatment ($C_{53}$) | 0.144 | 0.017 | 21 |
| | | Clean energy use ($C_{54}$) | 0.084 | 0.010 | 24 |
| | | Recycling/renewable capacity ($C_{55}$) | 0.197 | 0.023 | 17 |

## 4.3. Ranking Alliance Partners through Fuzzy Modified TOPSIS-AL Technique

The process of selecting green biopharmaceutical strategic alliance partners is complex and difficult. MCDM is one of the most effective ways to solve such problems because it is simple and fast to meet the needs of the managers to support the development of improved strategies. In view of the uncertainty of information and expert opinions, this study uses TOPSIS technique and fuzzy theory to strengthen the analytical model, and introduces the concept of aspiration level into the method. The efficiency of the model calculation is not affected by the number of alternatives. The fuzzy modified TOPSIS-AL technique analysis can be performed according to Section 3.2.

It is very difficult to directly convert an expert's subjective opinion into a general value. Therefore, linguistic variables are used to convert to triangular fuzzy numbers, which converts qualitative information into a useful solution for fuzzy numbers. The eight experts evaluated the performance of five strategic alliance partners based on the linguistic variables appearing in Table 2. The initial fuzzy decision matrix integrating the opinions of these eight experts is shown in Table 8. The concept of the aspiration level is introduced into TOPSIS. The triangular fuzzy numbers for the highest and lowest levels of the evaluation scale are (10, 10, 10) and (0, 0, 0). Using Equations (6) and (7), a normalized fuzzy decision matrix and a weighted normalized fuzzy decision matrix can be obtained, as shown in Tables 9 and 10. In practice, governments, enterprises and organizations will draw up a target and then move toward and make improvements to reach that goal. This goal correlates with the concept of the aspiration level proposed in this paper.

**Table 8.** Initial fuzzy decision matrix $\widetilde{X}$.

|       | $C_{11}$              | $C_{12}$              | $\ldots$ | $C_{55}$              |
|-------|-----------------------|-----------------------|----------|-----------------------|
| $A_1$ | (3.750, 4.750, 5.750) | (3.500, 4.500, 5.500) | $\ldots$ | (3.500, 4.500, 5.500) |
| $A_2$ | (3.500, 4.500, 5.500) | (4.250, 5.250, 6.250) | $\ldots$ | (4.750, 5.750, 6.750) |
| $A_3$ | (7.500, 8.500, 9.500) | (7.500, 8.500, 9.500) | $\ldots$ | (4.750, 5.750, 6.750) |
| $A_4$ | (5.500, 6.500, 7.500) | (5.250, 6.250, 7.250) | $\ldots$ | (4.500, 5.500, 6.500) |
| $A_5$ | (5.750, 6.750, 7.750) | (5.500, 6.500, 7.500) | $\ldots$ | (4.250, 5.250, 6.250) |

**Table 9.** Normalized fuzzy decision matrix $\widetilde{X}^*$.

|       | $C_{11}$              | $C_{12}$              | $\ldots$ | $C_{55}$              |
|-------|-----------------------|-----------------------|----------|-----------------------|
| $A_1$ | (0.375, 0.475, 0.575) | (0.350, 0.450, 0.550) | $\ldots$ | (0.350, 0.450, 0.550) |
| $A_2$ | (0.350, 0.450, 0.550) | (0.425, 0.525, 0.625) | $\ldots$ | (0.475, 0.575, 0.675) |
| $A_3$ | (0.750, 0.850, 0.950) | (0.750, 0.850, 0.950) | $\ldots$ | (0.475, 0.575, 0.675) |
| $A_4$ | (0.550, 0.650, 0.750) | (0.525, 0.625, 0.725) | $\ldots$ | (0.450, 0.550, 0.650) |
| $A_5$ | (0.575, 0.675, 0.775) | (0.550, 0.650, 0.750) | $\ldots$ | (0.425, 0.525, 0.625) |

**Table 10.** Weighted normalized fuzzy decision matrix $\widetilde{X}^{**}$.

|       | $C_{11}$              | $C_{12}$              | $\ldots$ | $C_{55}$              |
|-------|-----------------------|-----------------------|----------|-----------------------|
| $A_1$ | (0.016, 0.020, 0.024) | (0.021, 0.027, 0.033) | $\ldots$ | (0.008, 0.010, 0.013) |
| $A_2$ | (0.015, 0.019, 0.023) | (0.026, 0.032, 0.038) | $\ldots$ | (0.011, 0.013, 0.015) |
| $A_3$ | (0.031, 0.036, 0.040) | (0.045, 0.052, 0.058) | $\ldots$ | (0.011, 0.013, 0.015) |
| $A_4$ | (0.023, 0.027, 0.031) | (0.032, 0.038, 0.044) | $\ldots$ | (0.010, 0.013, 0.015) |
| $A_5$ | (0.024, 0.028, 0.032) | (0.033, 0.039, 0.045) | $\ldots$ | (0.010, 0.012, 0.014) |

According to Equations (8) to (12), the degree of separation of partner $A_i$ from PIS and NIS can be determined. It is confirmed that the degree of separation between the aspiration level and the positive ideal solution must be 0, and the degree of separation from the negative ideal solution must be 1. In contrast, the degree of separation between the worst level and the negative ideal solution is also 0, and the degree of separation from the positive ideal solution must be 1. The value of $CC_i$ ranges from $-1$ to 1. When the value of $CC_i$ is greater than 0, it indicates the group among all the evaluated partners with better performance because their evaluation results are closer to the expected value. Table 11 shows the results of the calculation combined with BWM and fuzzy modified TOPSIS-AL technique. The priority for partner selection is $A_3 > A_5 > A_4 > A_1 > A_2$. In order to visualize our evaluation results, examine Figure 2, which clearly illustrates the relative evaluation performance and room for improvement for partner $A_i$.

**Table 11.** The fuzzy modified TOPSIS-AL results and partner ranking.

|                   | $d_i^+$ | $d_i^-$ | $CC_i$   | **Rank** |
|-------------------|---------|---------|----------|----------|
| $A_1$             | 0.509   | 0.502   | $-0.005$ | 4        |
| $A_2$             | 0.525   | 0.485   | $-0.010$ | 5        |
| $A_3$             | 0.430   | 0.582   | 0.017    | 1        |
| $A_4$             | 0.480   | 0.531   | 0.003    | 3        |
| $A_5$             | 0.475   | 0.535   | 0.004    | 2        |
| Aspiration levels | 0       | 1       | 0.138    |          |
| Worst levels      | 1       | 0       | $-0.146$ |          |

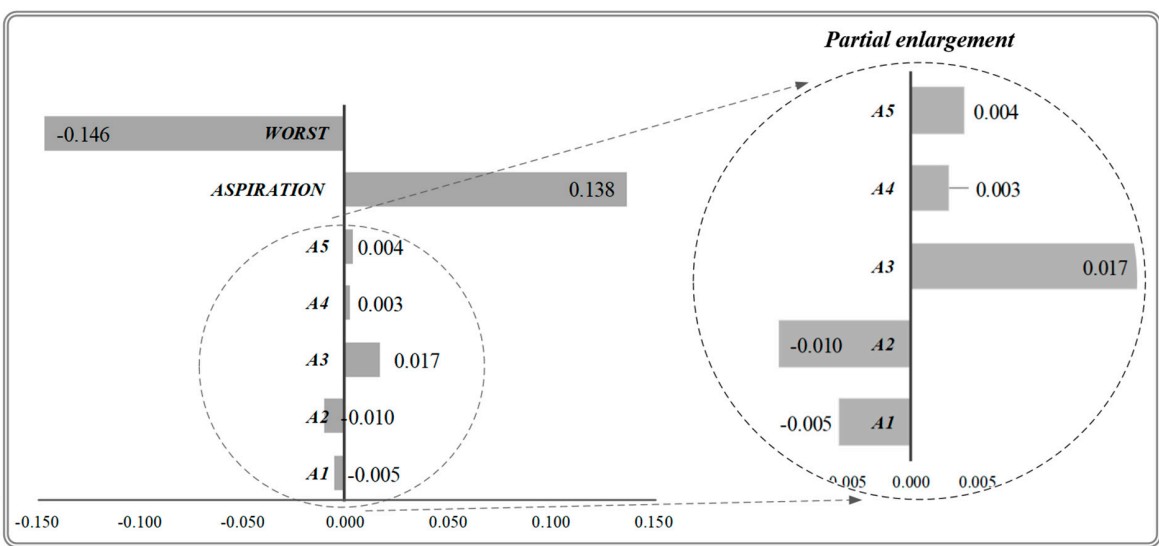

**Figure 2.** Closeness coefficients of the partners.

Figure 2 shows the distance between each partner and the aspiration level. Although $A_3$ ranks first in the rankings, the overall evaluation performance is still 0.121 (0.138–0.017) units away from the aspiration level, indicating that there is still much room for improvement. Traditional TOPSIS treats $A_3$ as an expected value, and this concept leads decision-makers to believe that $A_3$ does not need improvement. The model presented in this paper can overcome the above shortcomings and can provide more reliable management implications.

## 5. Discussion and Conclusions

The biopharmaceutical industry is one of the emerging high-tech industries, and advanced countries have invested huge sums of money to promote the industry. In order to improve the level of healthcare, biopharmaceutical-related products are constantly being developed. Compared to other manufacturing industries, the biopharmaceutical industry has a relatively high technical threshold, and most biopharmaceutical companies use strategic alliances to increase the competitiveness of the supply chain. According to the literature review, previous studies have rarely explored the evaluation framework of biopharmaceutical strategic alliance partners; in particular, the environmental protection criteria have not been established. This paper proposes an evaluation model for a green biopharmaceutical strategic alliance partner to bring a more complete evaluation framework and analysis method to the industry. First, through a large number of literature reviews and expert interviews, five dimensions and 25 criteria were established to establish an evaluation framework. Secondly, this paper uses BWM to obtain the criterion weight, which is an effective and reliable method for determining the weight in the MCDM problem, because it requires less pairwise comparisons and easy to obtain high consistency results. Finally, we improved TOPSIS, proposed by Kuo [30], by introducing the concepts of fuzzy theory and aspiration level to optimize the shortcomings of TOPSIS.

According to the BWM results of Table 7, the innovation capability ($D_2$) is the most important dimension based on the dimension level, with a weight value of up to 0.387. Product innovation capability is the most important competitiveness criterion for the biopharmaceutical industry. The government often uses patents as a basis for assessing the company's potential [43]. Zhang et al. [7] believe that R&D capabilities and patented technologies are key factors in the survival of the biopharmaceutical industry. Because of the long cycle of drug development and high investment costs, products can easily be imitated or even replaced without the support of patented technology. Their research echoes the results of our analysis. The two most important criteria are R&D capability ($C_{24}$) and core technical patent ($C_{21}$). Organizational management ($D_3$) is the second most important

dimension. The organizational cooperation of strategic alliances can be divided into organic coalitions, bureaucratic foundations, coalitions of intense interdependency, and reciprocal foundations. Regardless of the organizational approach, effective management mechanisms are needed to create mutually beneficial effects. Most of the strategic alliance partners hope to jointly create high-value brands and build customer loyalty through brand image to increase market share and revenue. We shared and fed back the results of the BWM analysis to all the experts, who say that this information can assist them in decision making in strategic alliances.

The proposed model provides a systematic analysis process that can completely evaluate and prioritize the partners. This study has confirmed that combining BWM with TOPSIS to analyze the strategic alliance partner problem should be an effective model. The calculation procedure proposed in this study optimizes TOPSIS. The results show that $A_3$ is currently the best performing strategic alliance partner. $A_3$ is a multinational food company with a turnover of NT$399.861 billion in 2017. The partner has a large sales channel, and in recent years, it wants to develop a pathway for the pharmaceutical industry, but lacks biopharmaceutical technology. Therefore, it is one of the members of the research case. Figure 2 shows that $A_3$ has relatively good performance compared to other partners. Based on the results of this evaluation, all partners can develop relevant improvement strategies to reach the aspiration level.

Although environmental protection ($D_5$) is not the most important dimension, it still has significant impact on the overall evaluation system. Since the green biopharmaceutical industry must pay attention to environmental protection, we explored whether the weight change of environmental protection ($D_5$) would affect the results of the overall evaluation system. Sensitivity analysis was used to verify that the partner's prioritization wold be changed significantly. The weight value of the environmental protection ($D_5$) was changed from 0.1 to 0.9, and the other criteria were weighted proportionally. Table 12 shows the ranking results of the nine test runs. Obviously, run five's partner ranking has changed. According to Figure 3, when environmental awareness becomes more and more important ($D_5$'s weight is getting higher and higher), the ranking of $A_1$ is getting higher and higher, indicating that $A_1$ environmental protection awareness is better than other partners. On the contrary, $A_5$ is a company that pays less attention to environmental protection. However, it is worth noting that when the weight of $D_5$ changes, it still does not affect the rankings of $A_2$ and $A_3$.

**Table 12.** The ranking results of the sensitivity analysis.

| Test | BWM | Run1 | Run2 | Run3 | Run4 | Run5 | Run6 | Run7 | Run8 | Run9 |
|------|-----|------|------|------|------|------|------|------|------|------|
| The weight of $D_5$ | 0.116 | 0.1 | 0.2 | 0.3 | 0.4 | 0.5 | 0.6 | 0.7 | 0.8 | 0.9 |
| $A_1$ | 4 | 4 | 4 | 4 | 4 | 3 | 3 | 2 | 2 | 2 |
| $A_2$ | 5 | 5 | 5 | 5 | 5 | 5 | 5 | 5 | 5 | 5 |
| $A_3$ | 1 | 1 | 1 | 1 | 1 | 1 | 1 | 1 | 1 | 1 |
| $A_4$ | 3 | 3 | 3 | 2 | 2 | 2 | 2 | 3 | 3 | 3 |
| $A_5$ | 2 | 2 | 2 | 3 | 3 | 4 | 4 | 4 | 4 | 4 |

In summary, the proposed evaluation model provides a systematic approach, a selection and evaluation tool for strategic alliance partners in the biotech pharmaceutical production industry. This effective soft computing method can reduce the subjectivity of management decisions. To the best of our knowledge, there has been no academic study exploring strategic alliances in the green biopharmaceutical industry. Our model integrates several state-of-the-art methods and considers various real-world situations, including the consideration of message uncertainty and the introduction of the concept of the aspiration level. Our results demonstrate the validity and reliability of the proposed model. Such a model would bring several benefits to the case company. It would: (i) make it easier to identify the most appropriate strategic alliance partner; (ii) provide a basis for improvement of the strategic partners; and (iii) help decision-makers be more systematic in the decision-making process.

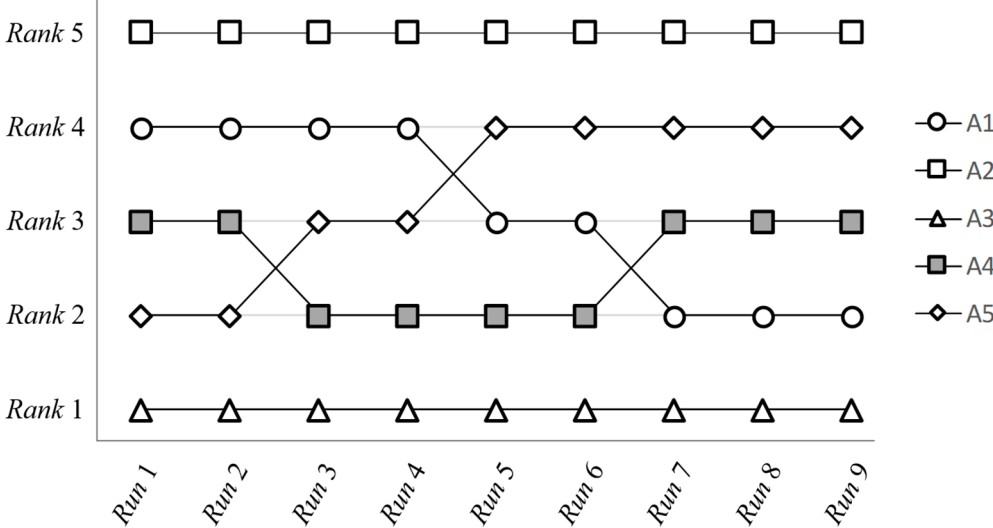

**Figure 3.** Diagram showing changes in partner ranking.

In addition, the results led to several new findings: (i) R&D capacity remains the most important condition for manufacturers; (ii) a biopharmaceutical production company must be supported by multiple invention patents to avoid being imitated by their competitors; (iii) sensitivity analysis reveals which partners are environmentally conscious, which will strongly influence sustainable development of the strategic alliances; (iv) organizational management is the second most important dimension for evaluation, with emphasis on the mutual assistance and mutual trust of partners in strategic alliances to jointly create an excellent brand image.

In the future, researchers can use different MCDM methods to evaluate partner performance, such as VIKOR, PROMETHEE, GRA, and DEA, etc. In addition, the quantitative data of actual enterprises can be further investigated to make the evaluation results more accurate.

**Author Contributions:** M.-H.C. analyzed the data, reviewed the literature, and wrote the paper. J.J.H.L. and H.-W.L. designed the research and co-wrote and revised the paper.

**Funding:** This research received no external funding.

**Conflicts of Interest:** All authors declare that they have no conflict of interests.

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
