# Peer review of "A Hybrid MCDM Model for Evaluating Strategic Alliance Partners in the Green Biopharmaceutical Industry"

_sustainability, doi:10.3390/su11154065_

Round 1
Reviewer 1 Report
I have gone through the paper this is interesting but a little bit complicated to the reader.
What is the motivation of the study regarding the research question and objectives?
Why do you believe this is a novel study through there are many studies have been investigating these strategic issues in the same or other industry?
The eight experts from the same company how could you avoid the biasedness of the study?
What types of question have been asked the expert?
Table 8, 9 and 10 are very complicated to the reader because of many dimensions.
What are the finding and contribution that we like to believe, the study has managerial implications because technological industry must need innovation capability?
No theoretical discussion didn't support the resources and innovation of the study.
Show the finding with figure 1.
Reviewer 2 Report
please clarify the following points.
What is green-biopharmaceutical industry?
Why studying strategic alliance is important in the green-biopharmaceutical industry? It needs further justification/explanation.
What is the basis of considering green biopharmaceutical industry energy intensive? What are the criteria?
Please provide the details of the process that led to the selection of the criteria presented in the figure 1.
Round 2
Reviewer 1 Report
The explanation extensively answered the questions of my mine.